# ADAPTIVE DECODING VIA LATENT PREFERENCE OPTIMIZATION

## ABSTRACT

During language model decoding, it is known that using higher temperature sampling gives more creative responses, while lower temperatures are more factually accurate. However, such models are commonly applied to general instruction following, which involves both creative and fact-seeking tasks, using a single fixed temperature across all examples and tokens. In this work, we introduce *Adaptive Decoding*, a layer added to the model to select the sampling temperature dynamically at inference time, at either the token or example level, in order to optimize performance. To learn its parameters we introduce *Latent Preference Optimization* (LPO), a general approach to train discrete latent variables such as choices of temperature. Our method outperforms all fixed decoding temperatures across a range of tasks that require different temperatures, including UltraFeedback, Creative Story Writing, and GSM8K.

## 1 INTRODUCTION

Large language models (LLMs) are powerful generalist models that can be used on a wide variety of tasks, ranging from fine-grained reasoning to open-ended creative writing (OpenAI, 2023; Dubey et al., 2024). Yet, early works showed that after training, the decoding method still has a large effect on performance across these tasks, leading to the proposal of various temperature sampling techniques (Holtzman et al., 2019; Welleck et al., 2019; Fan et al., 2018). In current LLMs, *temperature* (Ackley et al., 1985) is a key post-training parameter for generation. Temperature is used to scale the next token probabilities to be either more uniform or more sharp. Lower temperature leads to less creative, more factual generations, and higher temperature leads to more creative and original generations. Certain tasks, such as math problems or factual question answering, require the model to optimize accuracy of a single correct solution, and benefit from a low temperature, or greedy decoding (Shi et al., 2024). Others, like story generation, benefit from diverse and creative outputs, and a high decoding temperature. Intuitively, a complex task involving a number of these requirements might thus benefit from an adaptive temperature for different parts of its solution.

Existing LLM evaluation pipelines often rely on a fixed choice of temperature which is therefore suboptimal on some tasks, or else manual tuning is used to control the level of diversity in LLMs, which can be time-consuming, task-specific, and limited in its ability to adapt to changing requirements and prompts. To overcome this limitation, we introduce *Adaptive Decoding*, which consists of a new learnable layer, as well as a novel method to train it. The new learnable neural layer, which we call the ADAPTIVEDECODER, is added to the final layers of the transformer architecture, enabling the LLM to dynamically adjust its output diversity based on context (i.e, the task at hand). Specifically, the ADAPTIVEDECODER allows the model to select an ideal temperature for decoding the next token by adding a new decoder head attached to the last hidden state. We can either apply this at the example (sequence) level where a single temperature is predicted for all generated tokens, or the token level where a new temperature is predicted for each generated token.

Training the ADAPTIVEDECODER layer requires discrete optimization over latent variables (i.e., the choice of temperature). In order to make this feasible, we introduce a general method for such training, called Latent Preference Optimization (LPO). LPO involves sampling multiple responses from the model, where the ADAPTIVEDECODER layer will select temperatures (latent variables) that will affect the final token choices. Those responses are then evaluated by a reward model in order to build chosen and rejected preference pair examples. Given these pairs, we use the LPO loss to learn the optimal parameters of the ADAPTIVEDECODER layer for selecting temperatures during decoding.

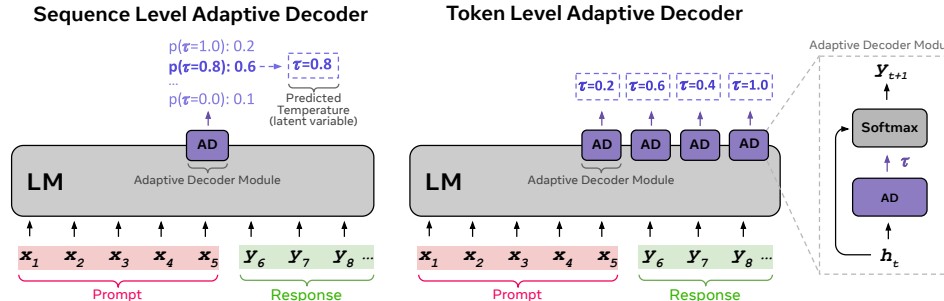

Figure 1: **The ADAPTIVEDECODER**. This learned module is added to the standard transformer in order to select decoding hyperparameters. It consists of a new decoder head attached to the last hidden state which assigns probabilities to different hyperparameter choices per token (right) or sequence (left), and the highest probability choice is selected in each case. This allows the LLM to select low temperatures for tokens requiring factual consistency, and higher temperatures for tasks requiring creativity and diversity. For the token level adaptive decoder, a different temperature can be selected for different parts of the response given a single instruction.

Our approach thus learns the hyperparameters of generating text across diverse tasks, allowing the model to balance exploration and exploitation in a task-aware manner.

To validate our method, we experiment on a diverse set of tasks, ranging from math reasoning to creative writing and general instruction following. We show that the decoder learns to select low temperatures for reasoning tasks like math, higher temperatures for creative writing, and somewhere in between for general prompts. We find that when the training data includes all types of tasks, the model adaptively adjusts the temperature to the ideal value for each task by conditioning output token temperature choices on the input context. This enables the ADAPTIVEDECODER to be incorporated as part of the standard post-training pipeline to produce a model that can adjust its diversity adaptively depending on the task at hand for general instruction following, and even use different decoding parameters within a single response for the best outcome. Additionally, our proposed approach is general, it could be potentially used to convert other hyperparameters (e.g. top-p, top-k) effectively into model parameters. Furthermore, we show that LPO is also a general tool to train discrete latent variables that can be used for other architecture choices that contain discrete decisions.

## 2 RELATED WORK

**Fixed Decoding Strategies.** Various methods have proposed different fixed decoding strategies that often depend on one or more hyperparameters. Holtzman et al. (2019) introduced nucleus sampling, Fan et al. (2018) introduced top-k sampling, and since then further sampling approaches have been proposed (Nguyen et al., 2024). Shi et al. (2024) showed that different decoding strategies work better for different tasks. Zhang et al. (2020) evaluates different decoding strategies including fixed temperature, top-k, and top-p. They find that when diversity is the priority, all methods perform similarly, but when quality is the priority, top-p is best. Using different temperatures for different tasks has also cemented itself as common wisdom for prompting LLMs (Achiam et al., 2023). Commercial LLM API guides even recommend using a low temperature for analytical tasks and a temperature close to 1.0 for creative tasks [1] .

**Adaptive Decoding.** Prior work studied the adaptive change of decoding parameters under different criteria such as based on the target task, approximate reward of the desired output, or the target likelihood score. Zhu et al. (2024) developed a decoding strategy that can adapt based on the probability distribution of the previous token while Zhu et al. (2023) uses a rule-based method to predict a temperature value for each token. Basu et al. (2020) uses the desired perplexity value to predict the optimal top-k hyper-parameter for each token. Finlayson et al. (2023) proposes basis-aware sampling that finds the optimal support over the next token distribution by addressing the softmax bottleneck issue. Unlike our approach, none of these methods learn to predict an adaptive decoding strategy, but rather use various test time heuristics. Li et al. (2024) propose a method to learn sample specific diversity values on dialogue tasks using an MSE loss, where the diversity

---

[1] https://docs.anthropic.com/en/api/complete, https://ai.google.dev/gemini-api/docs/text-generation

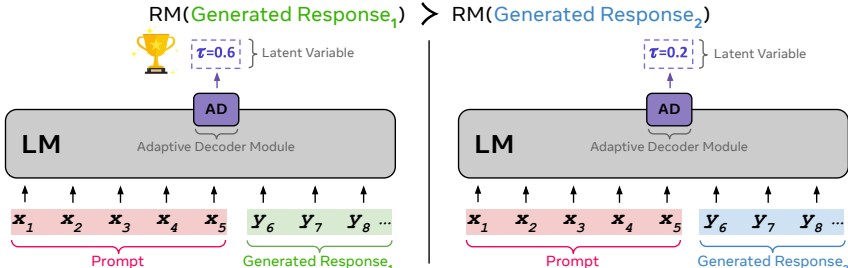

Figure 2: **Latent Preference Optimization (LPO) Training Mechanism.** We demonstrate how preference pairs are constructed for training the LPO loss (we show a Sequence-Level ADAPTIVEDECODER, but the procedure remains the same for Token-Level). Here we have N=2 generated response samples for a single prompt, and the Reward Model (RM) scores Response₁ better than Response₂. Therefore, we use $\tau = 0.6$ as the chosen temperature, and $\tau = 0.2$ as the rejected temperature, and then apply the loss to prefer the chosen temperature over the rejected one for the given context (prompt).

values are then mapped to temperatures using a mapping function. Zhang et al. (2024) dynamically select a temperature as a function of the entropy where the parameters of the function are treated as hyperparameters which they tune for each different task. Ad-hoc temperature prediction has been commonly used for calibration, as explored by Kumar & Sarawagi (2019) and Xie et al. (2024). Veličković et al. (2024) propose an adaptive temperature, where they vary the temperature for all softmax functions in the network depending on the entropy in the input coefficients. Xie et al. (2024) propose Adaptive Temperature Scaling to mitigate calibration errors, where they use a supervised loss which adapts targets depending on the correctness of the original model. To the best of our knowledge, we propose the first method to predict the temperature directly using preference optimization, allowing the model to learn task dependent temperatures at both the sequence and token levels.

**Preference Optimization.** Reinforcement Learning from Human Feedback (RLHF) has emerged as a major ingredient of LLM training (Ouyang et al., 2022). DPO (Rafailov et al., 2024) and other preference optimization methods (Xu et al., 2023; Meng et al., 2024) have significantly simplified the RLHF process. While many of these methods improve performance and generalization they can also negatively affect diversity and calibration (Achiam et al., 2023; Kirk et al., 2023). In particular, RLHF methods optimize the final reward which does not take diversity into account, so it has become common practice to add a KL regularization term to maintain some of the model's original diversity (Ziegler et al., 2019; Rafailov et al., 2024). To the best of our knowledge, our method is the first to use preference optimization for training latent variables instead of word tokens.

## 3 METHOD

The goal of our method is to make the language model itself choose an ideal temperature for generating tokens depending on the current context. To achieve this, we add a small differentiable module to an existing LLM that predicts a temperature value to be used for decoding word tokens, which we call the ADAPTIVEDECODER. For training an ADAPTIVEDECODER module, we develop a preference optimization method, LPO, that is designed for learning such hyperparameters. In the following subsections we describe the ADAPTIVEDECODER module and LPO loss in more detail.

### 3.1 ADAPTIVEDECODER MODULE

Here we introduce the ADAPTIVEDECODER module, which is a small neural network that can be attached on top of any existing LLM. It takes as an input latent representations of the last hidden layer and outputs a probability distribution over possible temperature choices. Let $\mathcal{M}$ be a transformer core (Vaswani, 2017) that maps a sequence of tokens $\{x_t\}$ to a latent representation, $h_t$, at the last layer. This latent representation is then usually converted to token probabilities using an un-embedding matrix $W$ followed by a softmax. Thus, a regular LLM generates the next token $x_{t+1}$ as follows:

$$h_t = \mathcal{M}(x_1, \ldots x_t), \quad x_{t+1} \sim \text{SOFTMAX}(Wh_t). \quad (1)$$

A fixed temperature value, $\tau$, can be used to scale the softmax distribution in the following way:

$$x_{t+1} \sim \text{SOFTMAX}(Wh_t/\tau), \quad (2)$$

where small temperature values (toward 0) make the distribution sharper, and high temperature values (toward 1) will result in the original distribution.

Adaptive Decoding works by predicting the optimal $\tau$ value for a specific input $\{x_t\}$. To use Adaptive Decoding, we also feed the LLM's hidden state $h_t$ to an ADAPTIVEDECODER module that maps it to a categorical probability distribution over a set of pre-defined temperature values $\tau_1, \ldots, \tau_K$:

$$P(\tau_k|h_t) = \text{ADAPTIVEDECODER}(h_t), \quad \text{where} \quad \sum_k P(\tau_k|h_t) = 1. \tag{3}$$

We can then straightforwardly make use of this distribution for generating a given output token, $x_{t+1}$, by selecting the $\tau$ with the highest probability, and then use that for decoding the next token:

$$\tau = \text{argmax}_{\tau_k} P(\tau_k|h_t), \quad x_{t+1} \sim \text{SOFTMAX}(Wh_t/\tau). \tag{4}$$

Alternatively, one can sample a *temperature* from the distribution and then sample a *token* with it:

$$\tau \sim P(\tau_k|h_t), \quad x_{t+1} \sim \text{SOFTMAX}(Wh_t/\tau). \tag{5}$$

This latter approach can also be written as a single sampling operation:

$$x_{t+1} \sim \sum_k P(\tau_k|h_t)\text{SOFTMAX}(Wh_t/\tau_k). \tag{6}$$

While the last two operations are identical, the second version will allow us to develop a new loss function for training as we will see in the next section.

Any neural network architecture can be used for the internals of the ADAPTIVEDECODER module, but we use a multi-layer perceptron (MLP) with a softmax output for simplicity (details in Section 4.1). Note that it is also straightforward to generalize the ADAPTIVEDECODER to other decoding hyperparameters such as top-k by simply modifying Equation 2 to the corresponding operation. In addition, $\mathcal{M}$ can be another neural model besides a transformer, such as a recurrent neural network.

## 3.2 TOKEN VS SEQUENCE LEVEL ADAPTIVEDECODER.

We propose two variants of the ADAPTIVEDECODER, as demonstrated in Figure 1. Let $\boldsymbol{x} = \{x_1, \ldots, x_T\}$ be the sequence of given input prompt tokens, and $\boldsymbol{y} = \{y_{T+1}, \ldots, y_{T'}\}$ be the generated response tokens. In the token level variant, ADAPTIVEDECODER$_{tok}$ (AD$_{seq}$), a temperature is predicted for each new token to be decoded. This is achieved by applying the ADAPTIVEDECODER at every step of generation and using the selected temperature for sampling the following token:

$$\tau_t \sim \text{ADAPTIVEDECODER}(h_t), \quad y_{t+1} \sim \text{SOFTMAX}(Wh_t/\tau_t) \text{ for } T \le t < T'. \tag{7}$$

Such fine-grained temperature adjustment allows the model to learn an individual temperature value for each token.

In the sequence level variant ADAPTIVEDECODER$_{seq}$ (AD$_{seq}$), a single temperature is predicted for the entire response. Unlike the token level, the ADAPTIVEDECODER module is used only once per input prompt, applied at the last token $x_T$ of the prompt to predict a temperature value $\tau$ to be used for the entire response generation:

$$\tau \sim \text{ADAPTIVEDECODER}(h_T), \quad y_{t+1} \sim \text{SOFTMAX}(Wh_t/\tau) \text{ for } T \le t < T'. \tag{8}$$

Such a coarse-grained temperature adjustment may be sufficient for most applications where the task requires either conciseness or creativity, but not both, and is still potentially much more flexible than the classical approach of choosing a single fixed temperature for all input prompts.

## 3.3 LATENT PREFERENCE OPTIMIZATION

To learn the ADAPTIVEDECODER parameters, we employ a preference optimization training where we generate multiple responses from the model and label some of them as *chosen* and others *rejected*. The overall goal of the training is to make the likelihood of generating chosen responses higher than the rejected ones, similar to the existing preference optimization methods such as DPO (Rafailov et al., 2024). However, those existing methods are designed to train token probabilities, not latent variables within the model. Thereby, we propose a generalization of DPO, which we call Latent

Preference Optimization (LPO), that is a general approach to train discrete latent variables, such as the choices of temperature[2].

To use LPO to learn optimal temperatures, we first generate multiple responses $\{\boldsymbol{y}^1, \ldots, \boldsymbol{y}^N\}$ for each prompt $\boldsymbol{x}$ by sampling temperatures from the ADAPTIVEDECODER output, which then affect how tokens are sampled. Let $\boldsymbol{\tau}^n = \{\tau^n_{T+1}, \ldots, \tau^n_{T'}\}$ be the temperatures used when generating tokens in response $\boldsymbol{y}^n = \{y^n_{T+1}, \ldots, y^n_{T'}\}$. The responses are then scored, either using an external reward model, or measuring the correctness of their answer, depending on the task. The highest and lowest scoring responses become our chosen and rejected response pair $(\boldsymbol{y}^c, \boldsymbol{y}^r)$. This process is depicted in Figure 2. Regular DPO training would optimize the token probabilities of these response pairs, but our goal is to learn the corresponding chosen and rejected temperature values $(\boldsymbol{\tau}^c, \boldsymbol{\tau}^r)$ that are used when sampling the response tokens. For this, there are multiple ways to adapt the DPO loss, which we outline below.

**Temperatures as tokens.** The simplest formulation is to treat the temperature selection just like another token. In this view, the model generates two tokens per step: a temperature token $\tau_t$ and a word token $y_t$. The temperature tokens have a different vocabulary, consisting of possible temperature values, but that does not complicate training. Similar to how the previous word token choice affects the next word token, the temperature token also affects the following word token probabilities. Since the model is generating a single sequence of "tokens", $\hat{\boldsymbol{y}}^n = (\boldsymbol{y}^n, \boldsymbol{\tau}^n)$, we can apply the usual DPO loss to this joint token sequence:

$$\mathcal{L}_{\mathrm{LPO}} = -\log \sigma \left[ \beta \log \frac{P(\hat{\boldsymbol{y}}^c)}{P_{\mathrm{ref}}(\hat{\boldsymbol{y}}^c)} - \beta \log \frac{P(\hat{\boldsymbol{y}}^r)}{P_{\mathrm{ref}}(\hat{\boldsymbol{y}}^r)} \right] = -\log \sigma \left[ \beta \log \frac{P(\boldsymbol{y}^c, \boldsymbol{\tau}^c)}{P_{\mathrm{ref}}(\boldsymbol{y}^c, \boldsymbol{\tau}^c)} - \beta \log \frac{P(\boldsymbol{y}^r, \boldsymbol{\tau}^r)}{P_{\mathrm{ref}}(\boldsymbol{y}^r, \boldsymbol{\tau}^r)} \right],$$

where $P_{\mathrm{ref}}$ are reference model probabilities. Since our reference model does not have an ADAPTIVEDECODER module, we omit it for the temperature tokens[3], and the loss therefore becomes:

$$\mathcal{L}_{\mathrm{LPO}} = -\log \sigma \left[ \beta \log \frac{P(\boldsymbol{y}^c)}{P_{\mathrm{ref}}(\boldsymbol{y}^c)} - \beta \log \frac{P(\boldsymbol{y}^r)}{P_{\mathrm{ref}}(\boldsymbol{y}^r)} + \beta \log P(\boldsymbol{\tau}^c) - \beta \log P(\boldsymbol{\tau}^r) \right]. \tag{9}$$

The advantage of this loss is that it takes into account both token and temperature probabilities, allowing for training both using a single loss. Here $\beta$ is a hyperparameter of DPO that controls the KL term.

**Temperatures as tokens (separate).** Like the previous formulation, we view the temperatures as tokens, but treat the word token generation as an external mechanism and focus only on the ADAPTIVEDECODER. In this view, the ADAPTIVEDECODER module generates a token $\tau_t$, which is a temperature value in this case, that is then fed to an external mechanism that generates the word token $y_t$. This framing makes things simpler because we have the ADAPTIVEDECODER generating two sequences of temperature values $(\boldsymbol{\tau}^c, \boldsymbol{\tau}^r)$ where one is preferred over the other. So we can directly apply the DPO loss with only the temperature tokens $\tau_t$:

$$\mathcal{L}_{\mathrm{LPO}} = -\log \sigma \left[ \beta \log P(\boldsymbol{\tau}^c) - \beta \log P(\boldsymbol{\tau}^r) \right]. \tag{10}$$

Again we omit the reference probabilities for the temperature tokens. This loss is simple and does not take account of token probabilities, but one can also use a separate DPO loss for the word tokens.

**Temperatures as latents.** In this version, we utilize the fact that the chosen and rejected labels are only conditioned on word tokens, and the temperature values that are used do not directly affect this ranking. The real objective we want to optimize is the probability of sampling chosen and rejected word sequences. Therefore, we treat the ADAPTIVEDECODER as an internal mechanism of the model and the temperature values as latent variables. This way, the model only outputs token probabilities like normal LLMs, but those probabilities are altered by the ADAPTIVEDECODER, as follows:

$$y_t \sim P'(y) = \sum_\tau P(y|\tau) P(\tau).$$

---

[2]While temperature is a continuous value, we are focusing on discrete temperature options in this paper. This also makes it easy to generalize our method to other discrete variables, such as top-k.

[3]This is the same as assuming the reference model has always uniform probabilities over possible temps.

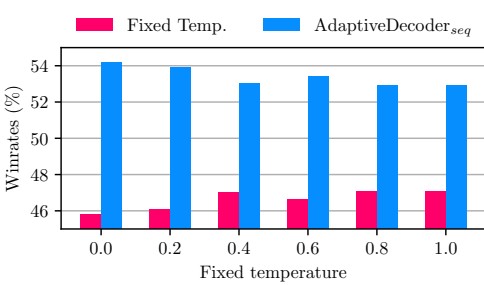 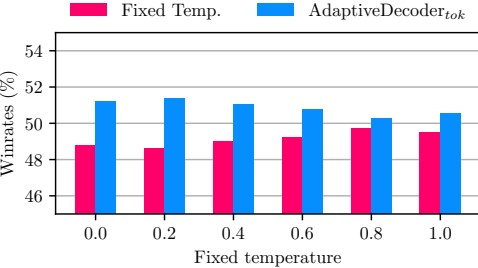

Figure 3: **UltraMathStories Results.** UltraMathStories is a superset of UltraFeedback, GSM8K, and Stories. The Adaptive Decoding models are trained on all 3 subtasks simultaneously. Winrates are shown as the average winrate across the test sets of the 3 subtasks in UltraMathStories. **(left)** ADAPTIVEDECODER$_{seq}$ vs Fixed Temperature Winrates. **(right)** ADAPTIVEDECODER$_{tok}$ vs Fixed Temperature Winrates. In both cases, Adaptive Decoding outperforms all fixed temperatures.

Table 1: **ADAPTIVEDECODER$_{seq}$ Predicted Temperatures ($\tau$) on UltraFeedback.** Examples of UltraFeedback test prompts where the ADAPTIVEDECODER$_{seq}$ model predicted $\tau \in \{0.0, 1.0\}$. Our model predicts the top prompt requires a factual deterministic response ($\tau = 0.0$), while the bottom prompt requires a creative, stochastic response ($\tau = 1.0$). More examples are shown in Table 13.

| Prompt | Predicted $\tau$ |
|---|---|
| Detailed Instructions: In this task, you are given a country name and you need to return the capital city of the given country\n Problem:Guinea-Bissau\n Solution: | 0.0 |
| Write a compelling short story about a bitter and intense rivalry between two individuals, where one must have an advantage in terms of their socioeconomic status or physical ability. The story must also incorporate a surprising twist that leads to an unforeseen outcome. | 1.0 |

Now we can apply the DPO loss to token probabilities where the temperature is marginalized out

$$\mathcal{L}_{\text{LPO}} = -\log \sigma \left[ \beta \log \frac{P'(\boldsymbol{y}^c)}{P'_{\text{ref}}(\boldsymbol{y}^c)} - \beta \log \frac{P'(\boldsymbol{y}^r)}{P'_{\text{ref}}(\boldsymbol{y}^r)} \right] = -\log \sigma \left[ \beta \sum_t \log \frac{P'(y_t^c)}{P'_{\text{ref}}(y_t^c)} - \beta \sum_t \log \frac{P'(y_t^r)}{P'_{\text{ref}}(y_t^r)} \right]$$

$$= -\log \sigma \left[ \beta \sum_t \log \frac{\sum_\tau P(y_t^c|\tau)P(\tau)}{\sum_\tau P_{\text{ref}}(y_t^c|\tau)P_{\text{ref}}(\tau)} - \beta \sum_t \log \frac{\sum_\tau P(y_t^r|\tau)P(\tau)}{\sum_\tau P_{\text{ref}}(y_t^r|\tau)P_{\text{ref}}(\tau)} \right]. \quad (11)$$

Note that the actual temperatures used in generation are irrelevant here, thus reducing the noise caused by sampling temperatures during training. The reference temperature probabilities $P_{\text{ref}}(\tau)$ are uniform if that is the initialization.

## 4 EXPERIMENTS

### 4.1 SETUP

For all experiments, we train an ADAPTIVEDECODER on top of a Llama 3.0-8B-Instruct model (Dubey et al., 2024). The ADAPTIVEDECODER module is a 3-layer MLP with hidden dimension 2048, and SiLU (Hendrycks & Gimpel, 2016) activations. We freeze the weights of the Llama model to better understand the effect of sampling temperature in isolation from finetuning the whole model. We use NVIDIA A100s for training and evaluation. For LPO training, by default we use the loss in Equation 10 for its simplicity, unless otherwise specified. During training, responses are generated using Equation 5 where temperatures are sampled, but we use greedy temperature selection at inference time using Equation 4, unless otherwise specified.

### 4.2 REWARD MODEL

We use an off-the-shelf reward model to score and rank generations for questions without ground-truth answers. We use the Armo Reward Model (ArmoRM) (Wang et al., 2024), a state-of-the-art model trained on diverse human preference data. ArmoRM outputs a scalar score, which we use during both

training and evaluation. At the time of our experiments, it ranked at the top of the RewardBench leaderboard (Lambert et al., 2024), with judgments closely aligned with human evaluations.

### 4.3 ULTRAMATHSTORIES

To test if, in realistic general instruction following settings, ADAPTIVEDECODER can learn to select different temperatures depending on the given prompt query. We thus deliberately consider a dataset that is a mixture of the following subtasks that require both formulaic, as well as creative responses:

- **Math (GSM8K).** When solving math reasoning problems, LLMs require greedy, or low-temperature sampling to produce accurate and reliable results (Kojima et al., 2022). The model should not deviate from high-likelihood tokens in this setting since factuality is crucial for finding the correct answer. GSM8K (Cobbe et al., 2021) is a common math reasoning dataset used to evaluate such capabilities. Since we have the ground truth answers, we use them to score responses to select training pairs, and for final test evaluation. We explain training details in Section A.2.

- **Creative Writing (Stories).** In contrast, when solving open-ended creative writing problems, LLMs benefit from high temperature sampling to write more interesting and original responses. We introduce a creative story writing task, which we call "Stories", to evaluate the creativity and coherence of a model on open ended prompts. We prompt the model to write a short story of a given title, where we use a language model to create the initial task titles. We use ArmoRM for scoring responses and use the highest and lowest scoring generations as preference pairs. See Section A.5 for more details on creating the dataset, and constructing the preference pairs.

- **General Instructions (UltraFeedback).** Finally, many real-world prompts lie somewhere in between structured reasoning and open-ended creativity or contain a mixture of both. We therefore consider the UltraFeedback (Cui et al., 2023) dataset, which covers a wide variety of real user prompts, ranging from rigid reasoning tasks to open-ended writing. We use the same ArmoRM for constructing and evaluating training preference pairs.

We combine 2,000 training samples from UltraFeedback, 1,000 training samples from GSM8K, and 1,000 training samples from the Stories dataset, and call it the "UltraMathStories" dataset. We train a single model on the preference pairs from this dataset to test if *Adaptive Decoding* can adapt to each subtask. We evaluate on each subtask's test set individually and take the average winrate across the 3 test sets. Further details of each subtask, including how the LPO pairs are created, are described in Section A.4. We experiment with both a sequence level and token level ADAPTIVEDECODER, and provide each with 6 temperature options: $\tau \in \{0.0, 0.2, 0.4, 0.6, 0.8, 1.0\}$.

**ADAPTIVEDECODER can learn to use the ideal temperature adapted for each subtask.** In Figure 3, we directly compare our method against fixed temperature decoding. The winrate in each subtask is computed (shown in Section B.1 and Section B.2) and their average is plotted. We observe the ADAPTIVEDECODER outperforming all of the fixed temperatures, which indicates that the decoder has learned to choose an ideal decoding temperature suited to each subtask. In fact, Figure 5 demonstrates this clearly with the predicted temperature distributions for each subtask. As expected, the ADAPTIVEDECODER predicts low temperature for math prompts (GSM8K), high temperature for creative writing prompts (Stories), and a mix of temperatures which are mostly in between for general instruction prompts (UltraFeedback). The latter has the biggest temperature variance, which makes sense given that it has more diverse prompts.

**Sequence-level vs. Token-level ADAPTIVEDECODER.** In this task, ADAPTIVEDECODER$_{seq}$ showed a stronger performance compared to ADAPTIVEDECODER$_{tok}$ as shown in Figure 3, even though both outperform fixed temperatures. There are several reasons why this can be the case. First, the subtasks in UltraMathStories themselves might not require fine-grained temperature adjustment. Secondly, learning a single temperature value per sample is much easier, thus likely to require fewer training samples (we only train on 4000 samples in total). However, we will explore the advantage of ADAPTIVEDECODER$_{tok}$ in subsequent sections.

### 4.4 CONSTRAINED CREATIVE WRITING (CONSTRAINEDSTORIES)

When given rigid instructions such as solving a math problem, the model needs to be greedy, but when given open-ended instructions such as writing a creative story, the model needs to be non-greedy. However, certain instructions can contain both rigid and open-ended instructions. We consider the

Table 2: **ADAPTIVEDECODER$_{tok}$ accuracy for majority voting ($N = 8$ samples) on the GSM8K dataset.** ADAPTIVEDECODER$_{tok}$ learns to assign appropriate temperatures at different parts of the generation which allows for more accurate sampled reasoning chains which results in a higher accuracy than using a single tuned temperature for the dataset. We also include the accuracy for $N = 1$ response, which underperforms majority voting.

| Decoding Method | Accuracy ↑ (N=8) | Accuracy ↑ (N=1) |
|---|---|---|
| Best Fixed Temperature | 87.46 | 81.59 |
| ADAPTIVEDECODER$_{tok}$ | 87.70 | 80.47 |
| ADAPTIVEDECODER$_{tok}$ (with $\tau$ 1.2) | **87.95** | 80.51 |

problem of constrained creative writing, which requires the model to be both greedy and non-greedy at different tokens of a single response.

We construct a dataset based on the Stories dataset from the previous subsection, and call it "ConstrainedStories". Similar to the Stories task, we prompt the model to write a creative story of a given title, but with an extra instruction saying that each sentence must start with a specific substring, "Ab" in this case. Intuitively, one would expect the ideal model should be greedy when generating the start of each sentence to satisfy the constraint, and non-greedy everywhere else for better creativity. The LPO preference pairs are created using both ArmoRM scores and constraint satisfaction rates. During evaluation, a higher constraint satisfaction wins, but ties are broken by the ArmoRM score. More details can be found in Section A.6.

**ADAPTIVEDECODER$_{tok}$ can learn to dynamically adjust the temperature at the token-level.** Figure 4 shows the winrates of ADAPTIVEDECODER$_{tok}$ compared to fixed temperature decoding. The ADAPTIVEDECODER$_{tok}$ always outperforms fixed temperature decoding. When a high fixed temperature is used on all tokens, it fails to follow the constraint, resulting in a low winrate. The greedy decoding performs well as it satisfies the constraint more often, but the story quality is lowered by the lack of diversity. Table 11 shows the individual winrates for constraint satisfaction and Armo score. As shown in Figure 7, the ADAPTIVEDECODER$_{tok}$ manages to have the best of both worlds. The average temperature for the first token of each sentence is $\tau = 0.21$, and the average temperature for all other tokens is $\tau = 0.55$. This shows that the model is mostly greedy on the constraint tokens (in order to generate an "Ab" word at the start of each sentence), and mostly non-greedy on all other tokens (in order to generate a creative and coherent story).Figure 6 shows an example of the ADAPTIVEDECODER$_{tok}$ predicted temperatures for a test sample prompt in this task.

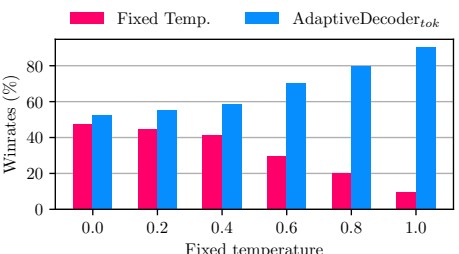

Figure 4: **Constrained Creative Writing (ConstrainedStories) Results.** Here we show a quantitative analysis of the ADAPTIVEDECODER on the constrained creative writing task, ConstrainedStories – ADAPTIVEDECODER$_{tok}$ winrates vs fixed temperatures. The high fixed temperatures perform worse because they fail to follow the constraint. Fixed greedy decoding works well at following the constraint, but ADAPTIVEDECODER$_{tok}$ outperforms it by using higher temperatures when possible.

## 4.5 MAJORITY VOTING

Wang et al. (2022) propose self-consistency, a method to improve the reliability of answers generated by language models by generating $N$ multiple independent reasoning chains and selecting the answer that appears most frequently. We explore whether the ADAPTIVEDECODER can learn to ascertain which parts of the reasoning chain should be sampled more stochastically and which should be decoded greedily. We explain further details about this experiment in Section A.3.

Generally, we find that increasing the fixed temperature above 1.0 can cause the LLM's generation to start to degrade and this can also hurt the performance of majority voting. However, the ADAPTIVEDECODER$_{tok}$ learns to assign temperatures appropriately and we observe that the higher temperature options help the model's performance, as shown in Table 2. This demonstrates that the

Table 3: **GSM8K accuracy training a sequence-level ADAPTIVEDECODER ($AD_{seq}$)** with different loss functions. We compare two different $\mathcal{L}_{LPO}$ loss functions, outlined in Section 3.3, as well as negative log likelihood, $\mathcal{L}_{NLL}$, trained on the chosen responses from preference pairs.

| | Fixed Temperature | | | ADAPTIVEDECODER$_{seq}$ | | |
|---|---|---|---|---|---|---|
| | $\tau = 0$ | $\tau = 0.6$ | $\tau = 1.0$ | $\mathcal{L}_{LPO}$ (Equation 10) | $\mathcal{L}_{LPO}$ (Equation 11) | $\mathcal{L}_{NLL}$ |
| | **81.59** | 79.15 | 78.32 | **81.59** | **81.59** | 78.92 |

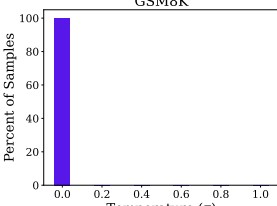 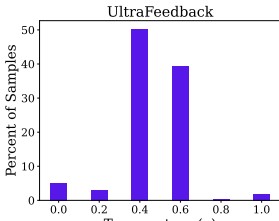 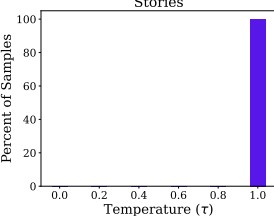

Figure 5: **ADAPTIVEDECODER$_{seq}$ predicted temperature distributions.** Distribution of predicted temperatures on the test set of each subtask in UltraMathStories. As expected, the model predicts low temperatures for GSM8K, high temperatures for Stories, and in between for UltraFeedback.

ADAPTIVEDECODER$_{tok}$ trained by LPO can result in a model that can perform well on both single responses (see Table 3 for single response accuracy) and majority voting setups at the same time.

## 4.6 ABLATIONS

**LPO Loss Type** As described in Section 3.3, there are several variations of the LPO loss that we can use. Here we compare two different LPO variants on the GSM8K math reasoning task: temperatures as tokens (separate) (Equation 10) and temperatures as latents (Equation 11). Table 3 shows the winrates of the ADAPTIVEDECODER$_{seq}$ model trained with the two different losses on the GSM8K math reasoning task. We see that both losses work well and match the greedy decoding (optimal) baseline. We also compare to a negative log-likelihood loss ($\mathcal{L}_{NLL}$), which is trained on only the chosen responses. This performs worse than both LPO losses since it tends to predict the most frequently chosen temperature, which is not necessarily the best temperature, as demonstrated in the training sample distribution plots in Figure 8.

**ADAPTIVEDECODER Temperature Selection** The objective of the ADAPTIVEDECODER is to predict the best temperature that is then used to scale the token probabilities for sampling a token. However, the LPO training learns a distribution of temperatures, not just a single value. Therefore, at inference time we can either greedily select the top temperature as in Equation 4, or sample from the temperature distribution following Equation 5, as we do for sampling from the token distribution. We compare these two different ways of selecting temperatures. Table 12 shows the winrates on UltraFeedback of the ADAPTIVEDECODER$_{seq}$ model trained on UltraMathStories (Section 4.3). Both methods outperform all fixed decoding temperatures, and we see a marginal difference between the two sampling methods.

## 5 CONCLUSION

As large language models continue to advance, users face important hyperparameter decisions, especially sampling temperature, which can balance exploration (generating creative and novel text) vs exploitation (generating conventional and factual text). In this paper, we introduce the ADAPTIVEDECODER, a neural module trained using our proposed Latent Preference Optimization (LPO) method, to dynamically predict decoding temperatures at inference time. Our experiments show that this adaptive approach consistently outperforms fixed temperatures, eliminating the need for manual tuning per task. In this work, we only experiment with adapting to the decoding temperature, however, Adaptive Decoding is general and can extend beyond temperature to other decoding hyperparameters such as top-p or top-k. By treating these hyperparameters as learnable parameters, our approach simplifies tuning and enables their optimization directly from the data.

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

# A  TASK DETAILS

## A.1  REDUCING N-GRAM REPETITIONS

We start with a simple first experiment where we know temperature choice matters. It is understood that LLMs are prone to erroneous repetitions, particularly when greedy decoding ($\tau$=0) is used (Holtzman et al., 2019). We therefore sought to validate whether the ADAPTIVEDECODER can learn to pick higher temperatures for specific tokens to avoid repeats. We use an ADAPTIVEDECODER$_{tok}$ and provide it with 5 temperature options: $\tau \in \{0.0, 0.1, 0.2, 0.4, 0.6\}$. We feed text from Wikitext-2 (Merity et al., 2016) to the model and ask it to complete it. We use 3-gram-repeats to rank the responses and create preference training pairs (see Section A.1 for details). We find that the ADAPTIVEDECODER$_{tok}$ effectively learns to reduce repeats by 42% compared to greedy decoding on the Wikitext-2 test set (Table 4). We also note that in around 94% of cases ADAPTIVEDECODER$_{tok}$ learns to pick a non-greedy temperature. This serves as a proof of concept that LPO can successfully optimize the temperature values in the right direction at the token level.

We use the Wikitext-2 benchmark. We use a 50 tokens prefix as the prompt, allowing the LLM to continue generating. After generating $N = 10$ completions per prompt, we rank these completions by 3-gram-repeats. We then constructed LPO preference pairs where the sequences with the lowest and highest 3-gram-repeats are selected as the 'chosen' and 'rejected' sequences respectively. We then use LPO to train the ADAPTIVEDECODER$_{tok}$ model.

Table 4: **Reducing Repeats using the ADAPTIVEDECODER.** We feed text from Wikitext-2 to the model and ask it to complete it. When completing a text, ADAPTIVEDECODER$_{tok}$ learns to avoid greedy decoding in order to reduce repeats. In 94% of samples, ADAPTIVEDECODER$_{tok}$ learns to pick a non-greedy temperature.

| Method | 3-gram-repeats ↓ | % of non-greedy |
|---|---|---|
| Greedy Decoding | 0.36% | 0% |
| ADAPTIVEDECODER$_{tok}$ | **0.22%** | 94% |

## A.2  MATH (GSM8K)

For this task, we use the GSM8K math reasoning dataset (Cobbe et al., 2021). We use chain-of-thought prompting (Wei et al., 2022), where the model is instructed to explain its reasoning before writing a final answer. We train on a random 1,000 sample subset of the full 7,473 samples. We evaluate on the full 1,319 test samples.

The LPO preference pairs for this dataset are constructed by generating $N = 16$ response samples per prompt, where each generation samples a temperature from the original ADAPTIVEDECODER distribution (roughly uniform), and then selecting a chosen and rejected sample based on the oracle GSM8K training labels.

We evaluate the performance of ADAPTIVEDECODER$_{seq}$ compared to 6 different fixed temperature decodings: $\tau = \{0.0, 0.2, 0.4, 0.6, 0.8, 1.0\}$. We measure the winrate of each test sample using the ground truth labels from the GSM8K test set. The winrate is computed by comparing the correctness of each method's response. If one method gets it correct and the other does not, the correct method gets awarded 1 point. If both methods generated a correct or incorrect response, then each method gets 0.5 points.

## A.3  MAJORITY VOTING

We first train a ADAPTIVEDECODER$_{tok}$ model on GSM8K to optimize the single response accuracy. We do this by sampling $N = 8$ responses and creating preference pairs using the ground-truth answers provided and apply LPO (Equation 10). Then we evaluate this model in a majority voting setting and compare it to the best fixed temperature decoding (tuned on the train set). We experiment with two categories of possible temperatures: $\{0.0, 0.4, 0.8, 1.0\}$ and $\{0.0, 0.4, 0.8, 1.0, 1.2\}$. To find the best-fixed temperature, we use line search. We find $\tau = 0.8$ works best for $N = 8$ while $\tau = 0.0$ works best for $N = 1$ sample.

### A.4 GENERAL INSTRUCTION FOLLOWING (ULTRAFEEDBACK)

The full UltraFeedback dataset contains 64k samples. We train on a random subset of 2,000 samples, and test on another random subset of 1,000 samples.

The training preference pairs for this dataset are constructed by generating $N = 16$ samples per prompt, where each generation samples a temperature from the original ADAPTIVEDECODER distribution (roughly uniform), and selecting a chosen and rejected sample using the best and worst Armo reward model (ArmoRM) (Wang et al., 2024) scores, respectively.

We measure the winrate of ADAPTIVEDECODER$_{seq}$ generations compared to each the 6 fixed temperature ($\tau=\{0.0, 0.2, 0.4, 0.6, 0.8, 1.0\}$) generations using ArmoRM scores.

### A.5 CREATIVE WRITING (STORIES)

For this task, we consider a simple creative writing task where the model is prompted to write a short story of a given title. Each prompt in this dataset has the following structure: "Write a short 200 word story with the following title.\n\nTitle:[TITLE]". We call this the "Stories" task. Each of the 1,000 training and test titles were generated with Llama3.0-70B.

We use the same method as UltraFeedback for constructing training preference pairs and evaluating.

### A.6 CONSTRAINED CREATIVE WRITING (CONSTRAINEDSTORIES)

Each sample has the following structure: "Write a creative and coherent story with the following title. You must begin each sentence with a word that starts with "Ab".\n\nTitle: [TITLE]".

The preference pairs are generated as follows. For each prompt, we first generate $N = 16$ response samples. To select the chosen response, we consider the top 4 ArmoRM scored responses, and then take the one of those that satisfy the constraint the best (has the highest percentage of sentences that start with "Ab"). Similarly, for the rejected response, we consider the bottom 4 ArmoRM scored responses and take the one of those that least satisfies the constraint.

Winrates are computed in the following way. If a response satisfies the constraint better (i.e., a higher percentage of "Ab" start sentences), then it wins. If there is a tie and both responses have the same constraint satisfaction rate, then it is decided by whichever response has a higher ArmoRM score, where the Armo reward model is run using the prompt without the constraint (i.e. "Write a creative and coherent story with the following title.\n\nTitle: [TITLE]").

## B WINRATE VALUES

### B.1 ULTRAMATHSTORIES ADAPTIVEDECODER$_{seq}$ WINRATE VALUES

Tables 5, 6 and, 7 show ADAPTIVEDECODER$_{seq}$ winrate values on each of the 3 UltraMathStories subtasks.

Table 5: **ADAPTIVEDECODER$_{seq}$ vs Fixed Temperatures Winrates on the *UltraFeedback* Task.**

| Fixed Temp | ADAPTIVEDECODER$_{seq}$ Winrate | Fixed Temp Winrate |
|---|---|---|
| $\tau = 0.0$ | 53.10 | 46.90 |
| $\tau = 0.2$ | 53.35 | 46.65 |
| $\tau = 0.4$ | 50.80 | 49.20 |
| $\tau = 0.6$ | 52.15 | 47.85 |
| $\tau = 0.8$ | 52.78 | 47.22 |
| $\tau = 1.0$ | 54.89 | 45.11 |

Table 6: **ADAPTIVEDECODER$_{seq}$ vs Fixed Temperatures Winrates on the Stories Task.**

| Fixed Temp | ADAPTIVEDECODER$_{seq}$ Winrate | Fixed Temp Winrate |
|---|---|---|
| $\tau = 0.0$ | 58.75 | 41.25 |
| $\tau = 0.2$ | 57.25 | 42.75 |
| $\tau = 0.4$ | 57.05 | 42.95 |
| $\tau = 0.6$ | 56.65 | 43.35 |
| $\tau = 0.8$ | 54.55 | 45.45 |
| $\tau = 1.0$ | 52.10 | 47.90 |

Table 7: **ADAPTIVEDECODER$_{seq}$ vs Fixed Temperatures Winrates on the *GSM8K* Task.**

| Fixed Temp | ADAPTIVEDECODER$_{seq}$ Winrate | Fixed Temp Winrate |
|---|---|---|
| $\tau = 0.0$ | 50.68 | 49.32 |
| $\tau = 0.2$ | 51.10 | 48.90 |
| $\tau = 0.4$ | 51.14 | 48.86 |
| $\tau = 0.6$ | 51.40 | 48.60 |
| $\tau = 0.8$ | 51.42 | 48.58 |
| $\tau = 1.0$ | 51.82 | 48.18 |

## B.2 ULTRAMATHSTORIES ADAPTIVEDECODER$_{tok}$ WINRATE VALUES

Tables 8, 9 and, 10 show ADAPTIVEDECODER$_{seq}$ winrate values on each of the 3 UltraMathStories subtasks.

Table 8: **ADAPTIVEDECODER$_{tok}$ vs Fixed Temperatures Winrates on the *UltraFeedback* Task.**

| Fixed Temp | ADAPTIVEDECODER$_{tok}$ Winrate | Fixed Temp Winrate |
|---|---|---|
| $\tau = 0.0$ | 49.60 | 50.40 |
| $\tau = 0.2$ | 50.70 | 49.30 |
| $\tau = 0.4$ | 48.75 | 51.25 |
| $\tau = 0.6$ | 49.60 | 50.40 |
| $\tau = 0.8$ | 49.25 | 50.75 |
| $\tau = 1.0$ | 52.75 | 47.25 |

Table 9: **ADAPTIVEDECODER$_{tok}$ vs Fixed Temperatures Winrates on the *Stories* Task.**

| Fixed Temp | ADAPTIVEDECODER$_{tok}$ Winrate | Fixed Temp Winrate |
|---|---|---|
| $\tau = 0.0$ | 54.40 | 45.60 |
| $\tau = 0.2$ | 53.40 | 46.60 |
| $\tau = 0.4$ | 54.20 | 45.80 |
| $\tau = 0.6$ | 52.30 | 47.70 |
| $\tau = 0.8$ | 51.10 | 48.90 |
| $\tau = 1.0$ | 47.25 | 52.75 |

Color Key: **$\tau$=0.0**, **$\tau$=0.2**, $\tau$=0.4, $\tau$=0.6, $\tau$=0.8, $\tau$=1.0

Prompt: "Write a creative and coherent story with the following title. You must begin each sentence with a word that starts with "Ab".\n\nTitle: The Village of the Blindfolded"

Response: "Abundant sunlight filtered through the trees as I made my way through the winding path that led to the Village of the Blindfolded. Abiding by the rules, I tied the blindfold around my eyes, just as the villagers did every day. Above the sound of chirping birds, I heard the soft murmur of whispers and the rustling of fabric as the villagers moved about. Abruptly, a gentle hand grasped my arm, guiding me towards the center of the village. Around us, the villagers stood in a circle, their faces expressionless behind their blindfolds. Above the gentle hum of the village, a soft melody drifted through the air, played on a lyre by a villager named Aria. Abiding by tradition, the villagers began to sway to the music, their footsteps quiet on the soft earth. Amidst the gentle movement, I felt a strange sense of freedom, as if the blindfold had freed me from the constraints of the world beyond the village. Above the music, a voice whispered in my ear, "Abandon your doubts, and you will find your true path." And so, I let go of my fears, and allowed the village to guide me towards the unknown.<|eot_id|>"

Figure 6: **ADAPTIVEDECODER$_{tok}$ predicted temperatures for Constrained Creative Story Writing.** We demonstrate an example of ADAPTIVEDECODER$_{tok}$ predicted temperatures ($\tau$) on the constrained creative story writing task for the prompt *"Write a creative and coherent story with the following title. You must begin each sentence with a word that starts with "Ab".\n\nTitle: The Village of the Blindfolded"*. We can see that the model is more greedy ($\tau$ close to 0.0) when generating the constraint tokens (All sentences must begin with words that start with "Ab"), and less greedy ($\tau$ close to 1.0) on all other tokens.

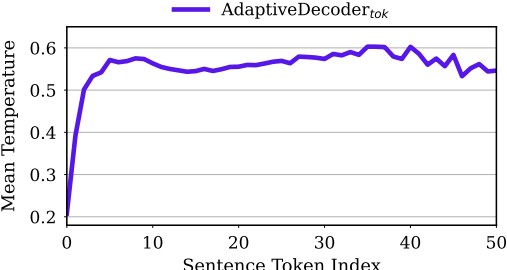

Figure 7: **Constrained Creative Writing (ConstrainedStories) Results.** Here we show a quantitative analysis of the ADAPTIVEDECODER on the constrained creative writing task, ConstrainedStories. Mean temperature predicted by the ADAPTIVEDECODER$_{tok}$ for the first 50 tokens of each sentence. This plot confirms our hypothesis that the first token of each sentence should be low temperature in order to follow the constraint, and all other tokens should be high temperature in order to write a good story. The average temperature for the first token is $\tau = 0.21$, and the average temperature for all other tokens is $\tau = 0.55$, showing a more greedy decoding for the constraint, and less greedy everywhere else.

Table 10: **ADAPTIVEDECODER$_{tok}$ vs Fixed Temperatures Winrates on the *GSM8K* Task.**

| Fixed Temp | ADAPTIVEDECODER$_{tok}$ Winrate | Fixed Temp Winrate |
|---|---|---|
| $\tau = 0.0$ | 49.66 | 50.34 |
| $\tau = 0.2$ | 50.08 | 49.92 |
| $\tau = 0.4$ | 50.11 | 49.89 |
| $\tau = 0.6$ | 50.38 | 49.62 |
| $\tau = 0.8$ | 50.49 | 49.51 |
| $\tau = 1.0$ | 51.55 | 48.45 |

Table 11: **ADAPTIVEDECODER$_{tok}$ Constrained Creative Writing Individual Winrates.** Here we show the individual winrates of the ADAPTIVEDECODER$_{tok}$ for both constraint following and ArmoRM score. The ADAPTIVEDECODER$_{tok}$ learns to follow the constraint better than all fixed temperatures, but as we compare to higher fixed temperatures, the story winrate goes down because it follows the constraint better.

| Fixed Temp | ADAPTIVEDECODER$_{tok}$ Constraint Winrate | ADAPTIVEDECODER$_{tok}$ ArmoRM Winrate | ADAPTIVEDECODER$_{tok}$ Avg Winrate |
|---|---|---|---|
| $\tau = 0.0$ | 50.95 | 52.55 | 51.75 |
| $\tau = 0.2$ | 53.70 | 49.50 | 51.60 |
| $\tau = 0.4$ | 58.05 | 48.25 | 53.15 |
| $\tau = 0.6$ | 68.05 | 41.05 | 54.55 |
| $\tau = 0.8$ | 77.85 | 36.45 | 57.15 |
| $\tau = 1.0$ | 87.80 | 31.50 | 59.65 |

# C  STATEMENTS

## C.1  ETHICS STATEMENT

We conform to the ICLR Code of Ethics at `https://iclr.cc/public/CodeOfEthics`

## C.2  REPRODUCIBILITY STATEMENT

We have provided all the necessary details to reproduce our results on publicly available benchmarks.

## C.3  LLM USAGE

We did not use LLMs for ideation or writing.

Table 12: **ADAPTIVEDECODER Temperature Selection Methods on UltraFeedback**. The ADAPTIVEDECODER outputs a distribution over temperature values $\tau$, so we can either sample $\tau$ from that distribution or greedily select the highest probability $\tau$. Here we show winrates against the fixed temperature decoding in the left column, using the ADAPTIVEDECODER$_{seq}$ model trained on Ultra-MathStories (Section 4.3). All the winrates are above 50%, which means the ADAPTIVEDECODER always outperforms the fixed temperature. Also, we do not observe a significant difference between the two temperature selection methods.

| | | Temperature Selection | |
| --- | --- | --- | --- |
| | | Greedy (Equation 4) | Sample (Equation 5) |
| | $\tau = 0.0$ | 53.10 | 52.80 |
| | $\tau = 0.2$ | 53.35 | 53.15 |
| **Fixed** | $\tau = 0.4$ | 50.80 | 51.75 |
| **Temp.** | $\tau = 0.6$ | 52.15 | 52.50 |
| | $\tau = 0.8$ | 52.78 | 53.65 |
| | $\tau = 1.0$ | 54.89 | 53.95 |

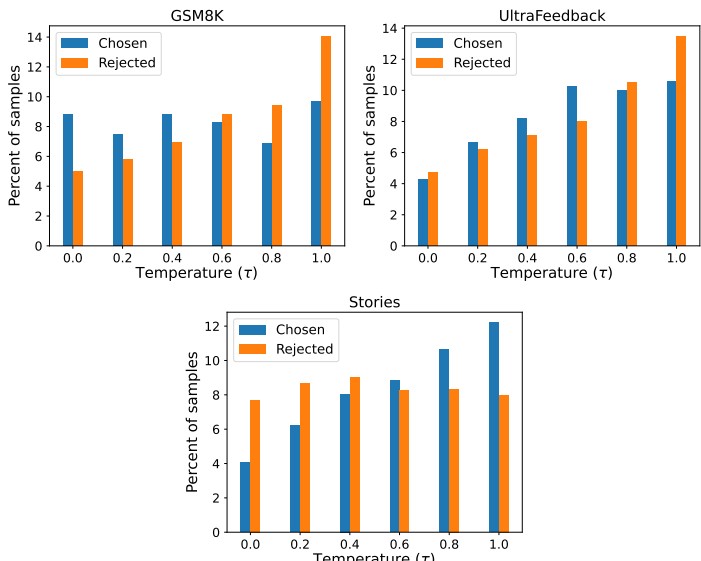

Figure 8: **ADAPTIVEDECODER$_{seq}$ Training Preference Distributions**. Here we show the percentage of samples in the training set that are chosen or rejected for each of the 6 different temperateure ($\tau$) values. The LPO loss uses both chosen and rejected responses, and the ratio of chosen to rejected is an important factor for learning the right temperature. A vanilla negative log-likelihood loss only uses the chosen responses, which leads to suboptimal temperature predictions since high-temperature values are the most chosen regardless of the task.

Table 13: Examples of **ADAPTIVEDECODER**$_{seq}$ **Predicted Temperatures ($\tau$) on UltraFeedback.** Here we show examples of UltraFeedback test prompts where the ADAPTIVEDECODER$_{seq}$ model predicted $\tau \in \{0.0, 1.0\}$. We can see that the $\tau = 0.0$ prompts require factual, deterministic responses, and the $\tau = 1.0$ prompts require creative, stochastic responses. This shows generalization outside of the GSM8K and Stories subtasks to specific prompts within UltraFeedback.

| Predicted $\tau = 0.0$ |
| --- |
| In this task, given a sentence in the English language, your task is to convert it into the Thai language. 
 Problem:The secondary principals' association head, Graham Young, said: T̈he NCEA system put pressure on schools to accumulate credits - and the easiest way to do that was to encourage students into internally assessed unit standards. 
 Solution: |
| You are given a math word problem and you are supposed to apply multiple mathematical operators like addition, subtraction, multiplication, or division on the numbers embedded in the text to answer the following question and then only report the final numerical answer. 

 Input: Consider Input: debby makes 67 pancakes . she adds blueberries to 20 of them and bananas to 24 of them . the rest are plain . how many plain pancakes are there ? |
| You have been tasked with arranging a group of travelers, each with different preferences and needs, onto various modes of transportation. There are four modes of transportation available: A, B, C, and D. Each mode has its own unique features and limitations. The travelers and their preferences are as follows: 
 1. Alice: Is afraid of flying and prefers to take mode C or D 
 2. Bob: Can only travel by mode A due to motion sickness 
 3. Charlie: Wants to take mode B because it has the shortest travel time 
 4. Dave: Needs to take mode D because he has a lot of luggage 
 5. Ellie: Wants to take mode A because she enjoys the scenic route 
 Your task is to assign each traveler to the mode of transportation that best suits their needs and preferences. Keep in mind that each mode of transportation can only accommodate a certain number of people, and some modes may have already reached their capacity. Can you solve this puzzle and successfully group the travelers onto their preferred modes of transportation?" |

| Predicted $\tau = 1.0$ |
| --- |
| Write a 70,000 word fantasy novel about a hidden world of magic and mythical creatures. The main character must be a human who discovers this world and becomes involved in a conflict between the magical creatures. The novel should have a fast-paced plot with plenty of action and suspense. The style should be descriptive and immersive, with detailed descriptions of the magical world and its inhabitants. The novel should also explore themes such as the nature of power and the importance of loyalty and friendship. |
| Write me a 1000 word ghost story in a campfire setting |
| Write a story about Ego Must, a prominent innovator with technology who leverages his vast wealth to communicate his views. However, despite being exceptionally smart he seems to not understand the basics when it comes to the 'us and them' problem that is at the root of a lot of human conflict. |

