# OpenReview forum: "Adaptive Decoding via Latent Preference Optimization"
_ICLR.cc/2026/Conference — Submitted to ICLR 2026_

### Official Review · Reviewer_NfwP · 2025-10-30

**Soundness:** 2
**Presentation:** 3
**Contribution:** 3
**Rating:** 4
**Confidence:** 3

**Summary:**

This paper introduces Adaptive Decoding, a method to dynamically select the sampling temperature during language model inference at either the token or sequence level. The authors propose a new module, the AdaptiveDecoder, which is trained using Latent Preference Optimization (LPO), a generalization of DPO for discrete latent variables. The method is evaluated on a mixed-task dataset (UltraMathStories) comprising math reasoning (GSM8K), creative writing (Stories), and general instruction following (UltraFeedback). Results show that adaptive temperature selection outperforms fixed-temperature baselines, and the model learns to assign low temperatures to factual tasks and high temperatures to creative ones. The method also demonstrates token-level adaptability in constrained creative writing.

**Strengths:**

- The author incorporates preference optimization into temperature generation and extends the approach to consider both sequence-level and token-level aspects.

- The authors evaluate on a diverse set of tasks requiring different temperature regimes, and show that the model learns to assign temperatures appropriately without manual tuning.

- The constrained creative writing task provides a compelling use case for token-level temperature control, showing that the model can learn to be greedy on constraint tokens and stochastic elsewhere.

**Weaknesses:**

- The paper only compares against a trival setting (fixed temperature), and does not include comparisons to existing adaptive temperature decoding methods [1-6].
- While the adaptive models consistently outperform fixed temperatures, the winrate improvements are often small (e.g., 51–55% winrates in Tables 5–10). In some cases (e.g., GSM8K), the gains are minimal, and the best fixed temperature (τ=0) is nearly as good as the adaptive model.
- Experiments are conducted only on Llama 3.0-8B-Instruct. There is no validation on other model families (e.g., Qwen, Gemma, Mistral) or model scales (e.g., 1B, 70B), which limits the generalizability claims.
- The paper focuses only on temperature, but does not explore other decoding hyperparameters (e.g., top-p, top-k), nor does it compare to other advanced sampling methods (e.g., nucleus sampling, typical decoding, contrastive decoding).

[1] Calibrating Language Models with Adaptive Temperature Scaling, EMNLP 2024
[2] To Cool or not to Cool? Temperature Network Meets Large Foundation Models via DRO, ICML 2024
[3] LLM can Achieve Self-Regulation via Hyperparameter Aware Generation, ACL 2024
[4] Improving open-ended text generation via adaptive decoding, ICML 2024
[5] Hot or Cold? Adaptive Temperature Sampling for Code Generation with Large Language Models, AAAI 2024
[6] Optimizing Temperature for Language Models with Multi-Sample Inference,  ICML 2025

**Questions:**

N/A

---

### Official Review · Reviewer_Hzzp · 2025-10-31

**Soundness:** 1
**Presentation:** 2
**Contribution:** 2
**Rating:** 2
**Confidence:** 4

**Summary:**

This paper introduces Adaptive Decoding, a lightweight module (3-layer MLP) attached to the final hidden states of a frozen LLM that predicts decoding temperature either per query or per token at inference time. Instead of using a single fixed temperature for every prompt, the module outputs a distribution over a small, discrete set of temperatures, the argmax is then used to scale the next-token distribution. To train this module, the authors propose Latent Preference Optimization, a preference-optimization objective adapted to discrete latent variables rather than word tokens. Experiments with Llama-3-8B-Instruct on several tasks, including instruction following, math, and creative writing, show that the proposed method outperforms the fixed temperature baseline.

**Strengths:**

* The motivation for task-dependent temperature is clear and addresses an important problem.


* The proposed MLP module is simple and easy to implement on top of existing LLMs.


* The paper includes some good ablation studies, such as those comparing different LPO variants.

**Weaknesses:**

* My first concern is the generalization ability of the proposed method. Since it still requires training on a mixture of datasets, how does it perform on out-of-distribution tasks? How does it handle tasks that require both high and low temperatures simultaneously? It would be helpful to include such evaluations.
* The evaluation is limited to the Llama-3-8B model and a fixed-temperature baseline. Because the paper focuses on adaptive temperature decoding, it’s important to compare and discuss prior work such as [1, 2, 3].
* The results do not clearly demonstrate the benefits of the proposed method on tasks such as GSM8K (Table 2).

[1] Nguyen, Minh Nhat, et al. "Turning up the heat: Min-p sampling for creative and coherent llm outputs." ICLR 2025.

[2] Tang, Chenxia, et al. "Top-n𝜎: Eliminating Noise in Logit Space for Robust Token Sampling of LLM." ACL 2025.

[3] Chang, Haw-Shiuan, et al. "REAL Sampling: Boosting Factuality and Diversity of Open-Ended Generation via Asymptotic Entropy." TACL 2025.

**Questions:**

I am not fully convinced by the idea of treating temperature as generated tokens, since temperature is a continuous value whereas typical word tokens are discrete. Could the authors clarify why this is reasonable?

---

### Official Review · Reviewer_e9aW · 2025-11-01

**Soundness:** 2
**Presentation:** 2
**Contribution:** 2
**Rating:** 2
**Confidence:** 3

**Summary:**

The paper addresses the issue that decoding strategies strongly affect the style, factuality, and creativity of LLM outputs. The commonly used sampling temperature controls this trade-off: low temperatures (τ≈0) yield deterministic and factual text, while high temperatures (τ≈1) produce diverse and creative generations. However, current LLMs typically use a fixed temperature during inference, which causes task mismatch (e.g., math vs. storytelling), inability to adjust across reasoning stages, and a lack of adaptability.

To solve this, the authors propose letting the model learn to choose its own temperature at inference time. They introduce two main innovations:
Adaptive Decoder (AD): a small plug-in MLP module appended to the LLM’s final layer that predicts the ideal temperature τ either at the sequence level (one τ per sample) or token level (a separate τ per token). This turns temperature from a static hyperparameter into a learnable, context-aware variable controlling output diversity.
Latent Preference Optimization (LPO): an extension of DPO/RLHF to optimize discrete latent variables such as temperature. For each prompt, the model generates multiple responses with different temperatures, a reward model (ArmoRM) ranks them, and LPO updates the Adaptive Decoder to favor temperatures leading to higher-scoring responses.

Experiments use Llama-3-8B-Instruct (frozen weights) with only the Adaptive Decoder trained on a mixed dataset UltraMathStories, combining GSM8K (low-τ math reasoning), Stories (high-τ creative writing), and UltraFeedback (mixed instructions). Results show that Adaptive Decoding outperforms all fixed-temperature baselines across subtasks, with predicted τ values aligning intuitively with task types.

**Strengths:**

The paper tackles a meaningful and timely topic — dynamically adjusting the decoding temperature of large language models, which directly affects their creativity, factuality, and overall controllability.

This is an underexplored yet practically important aspect of LLM inference. The proposed Latent Preference Optimization (LPO) framework and the accompanying Adaptive Decoder module are shown to be effective, achieving consistent improvements across diverse tasks.

The Adaptive Decoder itself is a lightweight, plug-and-play component that can be attached to existing models without retraining the base LLM, making the approach highly modular, easy to integrate, and potentially extensible to other decoding hyperparameters (e.g., top-p or top-k).

**Weaknesses:**

1. The base model (Llama-3-8B-Instruct, frozen) and training size (~4 k preference pairs) are modest. The results may not generalize to larger models or open-ended instructions.

2. The comparison setup is potentially unfair: the proposed AdaptiveDecoder introduces additional trainable parameters and receives preference-based supervision, while all fixed-temperature baselines are frozen and untrained. This makes it unclear whether the reported gains stem from adaptive temperature selection or simply from the extra learning signal.

3.  While the paper compares Adaptive Decoding against fixed-temperature decoding, all baselines are untrained models, whereas the proposed method introduces a trainable MLP head optimized via preference feedback. This makes the comparison asymmetric: improvements may stem from additional learning rather than genuine adaptivity. Moreover, the paper omits stronger adaptive decoding baselines such as entropy-based dynamic temperature (EDT; Zhang et al., 2024), rule-based adaptive sampling for code generation (Zhu et al., 2023; Li et al., 2024), and Adaptive Temperature Scaling (ATS; Xie et al., 2024) for calibration. Non-temperature adaptive sampling strategies like Mirostat (Basu et al., 2020) and Min-p sampling (Nguyen et al., 2024) also provide relevant points of comparison.

4. While the paper claims LPO as a "general" latent-variable preference optimizer, its actual implementation is very close to standard DPO with categorical latent tokens. There is no substantial theoretical analysis showing why LPO is distinct from treating temperature as another discrete action in DPO.

5. Significance tests are missing — many reported gains (≈ 2–3 %) might not be statistically robust.

**Questions:**

1. How sensitive is performance to the choice or quality of the reward model?

2. Since LPO requires generating N responses per prompt, what is the compute overhead compared to DPO or other RLHF algorithms?

---

### Official Review · Reviewer_WyS7 · 2025-11-02

**Soundness:** 4
**Presentation:** 4
**Contribution:** 4
**Rating:** 8
**Confidence:** 4

**Summary:**

Depending on the question you ask a model, you may want a different temperature. For creative tasks like story writing we may want a higher temperate to sample from more creative parts of the distribution and for mathematics we may want lower temperature to get a best first guess. In this paper, the authors present latent preference decoding where they add a new training head that decodes the temperature for generation.

**Strengths:**

- For a while I’ve wanted to see a paper out like this.
- This is a good approach that can be extendable to other decoding techniques: e.g., top-p, top-k, should the model reason for longer or answer, etc.
- This is a good paper.

**Weaknesses:**

Not too many weaknesses. See questions below.

Minor point, but for the bar plots (fig 3) usually you want to show them from zero otherwise small differences can be exaggerated. I’d appreciate some bootstrapped confidence intervals or the like for Figure 3. In general, this paper will benefit from confidence intervals in all tables.

**Questions:**

- Have the authors considered a good way of extending this for continuous temperature values? Instead of outputting a token we output a value.
- For mathematics tasks where we have a clear verifier we often care about best-of-n sampling — multiple independent decodings for the same prompt. I would have appreciated a longer analysis in section 4.5.
- I would love to see how this approach works for longer reasoning tasks. You could imagine that the token-based temperature sampling outperforms sequence based sampling where depending on what we’re generating we would want a higher or lower temperature.

---

### Meta-Review · Area_Chair_iaau · 2026-01-03

**Summary:**

The paper proposes an "Adaptive Decoding" method using Latent Preference Optimization to dynamically adjust temperature during inference. Reviewer WyS7 strongly supported the paper, praising the intuitive approach and extendability. However, Reviewers e9aW, Hzzp, and NfwP raised substantial methodological concerns. The primary issues were the lack of comparison against existing adaptive decoding baselines, the unfair comparison between a trained module and frozen baselines, and the limited evaluation scope (restricted to Llama-3-8B).

**Reviewer Concerns:**

**Addressed by Rebuttal:**
* None. (No rebuttal was submitted).

**Outstanding Concerns:**
* **Missing Baselines:** The paper failed to compare against relevant prior work on adaptive temperature/sampling. (Reviewers e9aW, Hzzp, NfwP)
* **Unfair Comparison:** The proposed method uses a trainable MLP while baselines are frozen, making it unclear if gains are due to adaptivity or extra training. (Reviewer e9aW)
* **Limited Evaluation:** Experiments were restricted to a single model size (8B) and family (Llama-3), raising generalization concerns. (Reviewers e9aW, NfwP)

**Reviewer Scores:**

* **Reviewer WyS7 (8 -> 6):** Likely to lower their score upon realizing the significant missing baselines and methodological issues pointed out by the other three reviewers.
* **Reviewer e9aW (2 -> 2):** The reviewer's concerns regarding the unfair comparison and missing baselines are fundamental flaws that remain unaddressed.
* **Reviewer Hzzp (2 -> 2):** The reviewer's concerns all remain outstanding.
* **Reviewer NfwP (4 -> 2):** Originally borderline, this reviewer would likely downgrade to a clear reject given the consensus among peers.

---

### Decision · Program_Chairs · 2026-01-26

Reject